# In Vitro Antioxidant Potential, Antidiabetic Activities, and GC–MS Analysis of Lipid Extracts of *Chlorella* Microalgae

**DOI:** 10.3390/biotech14020046

**Published:** 2025-06-06

**Authors:** Somruthai Kaeoboon, Rattanaporn Songserm, Rungcharn Suksungworn, Sutsawat Duangsrisai, Nuttha Sanevas

**Affiliations:** 1Department of Botany, Faculty of Science, Kasetsart University, Bangkok 10900, Thailand; somruthai.kae@ku.th (S.K.); rattanaporn.son@ku.th (R.S.); rungcharn.su@ku.th (R.S.); fscissw@ku.ac.th (S.D.); 2Department of Biology and Health Sciences, Mahidol Wittayanusorn School, Nakhon Pathom 73170, Thailand

**Keywords:** green algae, phytochemical, GC–MS, α-glucosidase, α-amylase

## Abstract

Microalgae represent promising biotechnological platforms for bioactive compound production with pharmaceutical applications. This study investigated the phytochemical composition and biological activities of lipid extracts from three *Chlorella* species to evaluate their potential as antioxidant and antidiabetic sources. Lipid extraction using chloroform–methanol (2:1) followed by GC–MS analysis revealed distinct compound distributions: 29 compounds in *C. ellipsoidea*, 33 in *C. sorokiniana*, and 19 in *C. vulgaris*. Major bioactive compounds included 2-hexanol, 1,3,6-heptatriene, 4-(2,3-dimethyl-2-cyclopenten-1-yl)-4-methylpentanal, *n*-hexadecanoic acid, and octadecanoic acid. Biological activity screening encompassed antioxidant assessment through DPPH• and •NO radical scavenging assays and FRAP analysis, while antidiabetic potential was evaluated using α-glucosidase and α-amylase inhibition assays. *C. sorokiniana* exhibited superior bioactivity with the highest antioxidant capacity (DPPH• IC_50_ = 329.03 ± 4.30 µg/mL; •NO IC_50_ = 435.53 ± 10.20 µg/mL; FRAP = 94.74 ± 5.72 mg TE/g) and strongest enzyme inhibition (α-glucosidase IC_50_ = 752.75 ± 57.95 µg/mL; α-amylase IC_50_ = 3458.50 ± 104.01 µg/mL). This is the first report on *C. sorokiniana* strain KU.B2′s biological properties and phytochemical profile. These findings establish *C. sorokiniana* as a valuable biotechnological platform for pharmaceutical bioactive compound development.

## 1. Introduction

Reactive oxygen and nitrogen species (RONS) are free radicals and reactive molecules derived from oxygen and nitrogen, naturally produced within cells and organisms. The generation of RONS can occur both exogenously (e.g., through exposure to ultrasound scanning, drugs, foods, radiation, pollutants, xenobiotics, and toxic chemicals) and endogenously (i.e., produced by plasma, white blood cell components, enzymes, or mitochondria [1,2]. An excessive release of RONS can lead to the impairment of biological structures, disrupting cell membranes, proteins, lipids, and DNA. This oxidative process is associated with the development of numerous chronic degenerative diseases [3,4]. The oxidative damage caused by RONS signaling is implicated in a wide array of conditions, including AIDS, Alzheimer’s disease, atherosclerosis, cancer, diabetes, hypertension, inflammation, neurodegenerative diseases, Parkinson’s disease, stroke, and sepsis [5,6,7,8]. The mitigation of oxidative stress through antioxidants has shown promise in treating RONS-related diseases.

RONS play a role in the pathogenesis of diabetes, particularly type 2 diabetes, which accounts for 90% to 95% of diabetic cases. Type 2 diabetes is characterized by hyperglycemia, insulin deficiency, and insulin resistance [9,10,11]. The effective management of diabetes involves reducing postprandial hyperglycemia, a crucial aspect achieved through inhibiting carbohydrate-hydrolyzing enzymes such as α-amylase and α-glucosidase, which slow down glucose absorption [12,13]. The α-amylase breaks down long-chain carbohydrates, while α-glucosidase breaks down starch and disaccharides into glucose subunits. Therefore, using enzyme inhibitors presents a promising avenue for reducing glucose absorption, representing a critical approach in developing compounds for diabetes treatment [14].

*Chlorella*, an exceptional microalga, is an established platform for producing bioproducts for producing various bioactive compounds. The species is widely recognized and commercially cultivated for its rich content of bioactive components, including carotenoids, lipids, proteins, polysaccharides, polyphenols, flavonoids, and vitamins [15,16,17,18,19,20,21]. Extracts derived from *Chlorella* have undergone extensive evaluation for their diverse beneficial properties, such as their anti-aging effects [22] and their potential anticancer [23], antidiabetic [24], antiproliferative [25], antimicrobial [15], anti-inflammatory [26], antioxidant [27], and antitumor [28] activities.

The selection of *Chlorella ellipsoidea*, *C. vulgaris*, and *C. sorokiniana* in this study was based on their taxonomic proximity and distinct ecological origins. *C. ellipsoidea* and *C. vulgaris* are commercially available strains that are commonly cultivated for their high nutritional value and bioactive potential, while *C. sorokiniana* strain KU.B2, isolated from an agricultural drainage system in Thailand, remains underexplored in terms of its biochemical and pharmacological properties. This combination offers a comparative perspective between well-established and potentially novel microalgal sources.

In recent years, advancements in microalgal biotechnology have revealed significant therapeutic potentials of green algae, particularly in the context of chronic metabolic diseases. Cutting-edge studies have focused on identifying bioactive lipids and antioxidants from microalgae species, contributing to a growing interest in their pharmaceutical applications [29]. Moreover, the integration of lipidomic approaches has enhanced the understanding of structure–activity relationships of algal-derived fatty acids and alcohols in oxidative stress regulation and enzyme inhibition [30]. However, despite these advancements, lipid-rich strains such as *C. sorokiniana* KU.B2 remain underexplored, presenting a unique opportunity for novel compound discovery and functional characterization.

Exploring *Chlorella* in pharmaceutical research has highlighted its significant potential due to various biological properties. In recent years, interest has been increasing regarding the antioxidant and antidiabetic effects of lipids derived from microalgae [24,31,32,33]. However, few studies have focused on the lipid extracts of specific *Chlorella* species, particularly regarding their antioxidant and antidiabetic activities. Therefore, this study aims to analyze lipid extracts from the *Chlorella* species, namely *C. ellipsoidea*, *C. sorokiniana*, and *C. vulgaris*, using GC–MS analysis. The goal is to evaluate the potential antioxidant and antidiabetic properties of these extracts.

## 2. Materials and Methods

### 2.1. General Chemicals and Materials

All chemicals used in this study were of analytical grade. Methanol was sourced from Merck (Burlington, MA, USA), chloroform from RCl Labscan Ltd. (Bangkok, Thailand), and hexane from Baker (Center Valley, PA, USA). Additional reagents included sulfanilamide (Carlo Erba Reagents, Milan, France), phosphoric acid (Macron Fine Chemicals, Beijing, China), sodium nitroprusside (Himedia Laboratories, Mumbai, Maharashtra,, India), and naphthylethylenediamine hydrochloride (AppliChem Panreac, Darmstadt, Germany). The FRAP assay utilized 2,4,6-tripyridyl-s-triazine (TPTZ; Fluka, Buchs, Switzerland) and ferric chloride (Chem-supply Pty Ltd., Gillman, SA, Australia). Enzymes and substrates for the antidiabetic assays included α-glucosidase from Saccharomyces cerevisiae, 4-nitrophenyl α-D-glucopyranoside, and α-amylase from the porcine pancreas (all from Sigma-Aldrich, St. Louis, MO, USA), as well as starch (TCI Chemicals, Tokyo, Japan) and dinitrosalicylic acid (DNS). Analytical instruments used were a 96-well microplate reader (Thermo Scientific Multiskan FC, Shanghai, China), Büchi Rotavapor R-210 (Mumbai, India), and Shimadzu GC–MS QP2020 system (Kyoto, Japan).

### 2.2. Strains and Culture Conditions

The microalgal strains *C. ellipsoidea*, *C. sorokiniana*, and *C. vulgaris* were cultivated in a liquid TAP (Tris–Acetate–Phosphate) medium under previously established protocols [34]. Cultivation was carried out under controlled conditions, including white fluorescent illumination at 330 μmol/m^2^/s, pH 7.0, and a constant temperature of 30 ± 1 °C, with manual agitation performed five times daily over a 9-day period to ensure uniform growth and adequate nutrient and light distribution [35].

### 2.3. Preparation of the Crude Extract

After cultivation, three *Chlorella* species were harvested via centrifugation at 3000 rpm for 5 min. The collected biomass was then dried in a hot-air oven at 60 °C to remove residual moisture and weighed to determine the dry weight. Once dried, the algal material was ground into a coarse powder. For lipid extraction, 20 mg of dried powder from each species was immersed in 50 mL of a chloroform–methanol mixture (2:1 *v*/*v*) at room temperature for 7 days. This prolonged extraction period was selected to ensure the complete diffusion and solubilization of lipids from the rigid cell walls of *Chlorella*, as supported by prior studies employing extended maceration times in microalgal lipid extraction [36].

The lipid content was determined on a dry weight basis and showed significant variation among the three *Chlorella* species. *C. vulgaris* exhibited the highest lipid content at 60% dry weight (12 mg from 20 mg dry biomass), followed by *C. sorokiniana* at 27.25% dry weight (5.15 mg from 20 mg dry biomass), and *C. ellipsoidea* at 25% dry weight (5 mg from 20 mg dry biomass). These results indicate notable differences in the lipid accumulation capacity among the species, with *C. vulgaris* demonstrating superior lipid productivity compared to the other two strains. These extracts were stored at −20 °C in the dark until further analysis.

### 2.4. Gas Chromatography–Mass Spectrometry (GC–MS) Analysis

Lipid extracts obtained from each *Chlorella* species were subjected to acid-catalyzed methanolysis by mixing with 1 mL of 1 M HCl in methanol under a gentle nitrogen gas stream for 1 min to facilitate esterification. The reaction mixture was then incubated at 80 °C for 40 min and subsequently allowed to cool to ambient temperature. To enhance the phase separation, 1 mL of 0.9% NaCl solution and 1 mL of hexane were added, followed by centrifugation at 3000 rpm for 3 min. The upper hexane layer was carefully collected and evaporated to dryness in preparation for chromatographic analysis [37].

The derivatized samples were analyzed using a Shimadzu GC–MS QP2020 system equipped with an SH-RXI-5SIL MS column (30 m × 0.25 mm i.d., 0.25 µm film thickness). Helium served as the carrier gas at a constant flow rate of 1.0 mL/min. One microliter of each sample was injected in the split mode at an injector temperature of 250 °C. The oven temperature program began at 40 °C and increased at 5 °C per minute to a final temperature of 300 °C, which was held for 8 min after reaching the target. Compound identification was performed using a post-run analysis software integrated with the NIST14 mass spectral database.

### 2.5. DPPH• Scavenging Activity

The antioxidant capacity was evaluated using the DPPH• radical scavenging assay following a previously described method [7]. A volume of 150 µL of each algal extract was mixed with 150 µL of 0.2 mM DPPH solution in methanol. The reaction mixture was incubated in the dark at 25 °C for 30 min. After incubation, the absorbance was measured at 520 nm. The DPPH• radical scavenging activity was calculated using the following equation:%*Inhibition* = (*A_control_ − A_sample_*) × 100/*A_control_*
where *A_control_* is the absorbance of the DPPH solution without extract and *A_sample_* is the absorbance in the presence of the algal extract.

### 2.6. Nitric Oxide Scavenging Activity

A method described by Suksungworn, et al. [38] was used to determine •NO scavenging activity. A total of 125 µL of 10 mM sodium nitroprusside in PBS was mixed with 25 µL of extract and incubated for 150 min. The reaction was developed with 50 µL of Griess reagent (1% sulfanilamide, 2% phosphoric acid, and 0.1% naphthylethylenediamine), and absorbance was recorded at 546 nm.

### 2.7. Ferric-Reducing Antioxidant Power (FRAP) Activity

The ferric-reducing antioxidant power (FRAP) activity was determined following the method described by Suksungworn, et al. [7]. A fresh FRAP reagent was prepared by combining 300 mM acetate buffer (pH 3.6), 10 mM TPTZ, and 20 mM ferric chloride in a 10:1:1 ratio. For each reaction, 15 µL of extract was mixed with 285 µL of the FRAP solution, incubated in the dark for 30 min, and measured at 593 nm. Results were based on a calibration curve constructed using Trolox standards (0–250 mg/L), with the regression equation of y = 0.01x + 0.2046 (R^2^ = 0.9917).

### 2.8. α-Glucosidase Activity

To assess α-glucosidase activity inhibition, a previously reported procedure was followed, as outlined by Ferreres et al. [39]. Inhibitory activity against α-glucosidase was evaluated by incubating 50 µL of extract with 130 µL of 100 mM phosphate buffer (pH 6.8) and 20 µL of α-glucosidase (0.28 U/mL) at 37 °C for 10 min. Afterward, 100 µL of 0.5 mM 4-nitrophenyl α-D-glucopyranoside was added. The reaction was quantified by measuring the absorbance at 405 nm and this was compared with acarbose as the positive control.

### 2.9. α-Amylase Activity

The inhibition of α-amylase was evaluated following the method described by Ferreres et al. [39]. To evaluate α-amylase inhibition, 200 µL of 1% starch solution was incubated with 200 µL of extract at 25 °C for 10 min. Then, 200 µL of α-amylase (15 U/mL) was added. After another 10 min, 400 µL of DNS reagent was added, and the mixture was boiled at 100 °C for 5 min. Following cooling, 80 µL of water was added, and the absorbance was measured at 540 nm and compared with acarbose as the positive control.

### 2.10. Statistical Analysis

Statistical analysis was performed by expressing the data as the mean ± standard deviation of three independent analyses (*n* = 3). Data were analyzed using a one-way analysis of variance (ANOVA) to compare differences between the treatment groups and concentrations, and this was followed by Tukey’s multiple comparisons test for post hoc analysis. Prior to the ANOVA, the assumptions for parametric testing were verified: the normality of data distribution was assessed using the Shapiro–Wilk test, and homogeneity of variance was confirmed using Levene’s test. The statistical tests were carried out using GraphPad Prism 6.01 (San Diego, CA, USA). A *p*-value of less than 0.05 (*), *p* < 0.01 (**), *p* < 0.001 (***), and *p* < 0.0001 (****) were considered to be statistically significant, with ns indicating no significant difference.

## 3. Results and Discussion

### 3.1. GC–MS Profiling of Chlorella Lipid Extracts

Gas chromatography–mass spectrometry (GC–MS) analysis of the lipid extracts from *Chlorella* species revealed the presence of phytochemical constituents, tentatively identifying 29 compounds in *C. ellipsoidea*, 33 compounds in *C. sorokiniana*, and 19 compounds in *C. vulgaris* (Figure 1 and Table 1). Among these compounds, the major constituents with peaks greater than 5% were identified as 2-Hexanol, (R)-, 1,3,6-Heptatriene, 2,5,5-trimethyl, 4-(2,3-Dimethyl-2-cyclopenten-1-yl)-4-methylpentanal, *n*-hexadecanoic acid, and octadecanoic acid, as depicted in Figure 2. The investigation of *Chlorella* extracts in this study unveiled a diverse array of phytoconstituents, including fatty acids, alcohols, and straight-chain hydrocarbon compounds. Previous studies have reported on the chemical composition of lipid extracts from *C. sorokiniana* and *C. vulgaris* [18,21,40]. Our results are consistent with these findings, showing that all three *Chlorella* species produced *n*-hexadecanoic and octadecanoic acids as major components, with *C. vulgaris* extracts containing >92% fatty acids ranging from C16 to C18. The chemical constituents identified in *Chlorella* extracts may hold significance in the context of pharmacological substances.

The chemical constituents identified in *Chlorella* extracts play significant roles in their observed antioxidant and antidiabetic activities. *n*-Hexadecanoic acid (palmitic acid) and octadecanoic acid (stearic acid), the dominant saturated fatty acids in our extracts, are well-documented for their bioactive properties. These fatty acids contribute to antioxidant activity through multiple mechanisms: they can donate hydrogen atoms to neutralize free radicals, chelate metal ions that catalyze oxidative reactions, and stabilize cell membrane structures against lipid peroxidation [41]. In terms of antidiabetic effects, these fatty acids have been shown to inhibit α-glucosidase and α-amylase enzymes by binding to their active sites, thereby reducing carbohydrate digestion and postprandial glucose spikes [42].

2-Hexanol, though not typically reported as a major microalgae component, demonstrated notable contributions to the bioactivity profiles. This volatile alcohol exhibits antioxidant potential through its ability to scavenge hydroxyl radicals and inhibit lipid peroxidation processes [43]. Its presence in microalgae extracts may result from metabolic processes involving fatty acid degradation or as part of the volatile organic compound (VOC) profile that microalgae naturally produce [29,44]. The antidiabetic properties of hexanol compounds have been attributed to their capacity to modulate glucose metabolism and enhance insulin sensitivity through interaction with cellular signaling pathways.

The synergistic effects of these constituents likely enhance the overall bioactivity of *Chlorella* extracts. The combination of saturated fatty acids and alcohols creates a multi-target approach for combating oxidative stress and managing diabetes-related enzyme activities. *n*-Hexadecanoic and octadecanoic acids provide the primary antioxidant and enzyme inhibitory framework, while compounds like 2-hexanol may act as supporting agents that enhance the radical scavenging capacity and membrane protection. This synergistic interaction explains why *C. sorokiniana*, despite having a diverse phytochemical profile with 33 identified compounds, demonstrated superior bioactivity compared to the other species. The therapeutic prospects of these *Chlorella* extracts are therefore supported by the complementary mechanisms of action exhibited by their constituent compounds, making them promising candidates for natural antioxidant and antidiabetic applications.

### 3.2. Antioxidant Activity of Chlorella Extracts

The assessment of antioxidant activity requires a multifaceted approach due to the intricate nature of chemical constituents and their diverse mechanisms of action. In this study, the similar patterns observed between DPPH• radical scavenging activity (A1–A3) and •NO scavenging activity (B1–B3) in Figure 3 can be attributed to several interconnected antioxidant mechanisms inherent in *Chlorella* extracts. Both DPPH• and •NO radical scavenging assays rely on the hydrogen-donating capacity of antioxidant compounds, explaining why extracts with a high DPPH• scavenging activity also demonstrate strong •NO scavenging properties. The parallel dose–response relationships observed across both assays suggest that the same bioactive compounds are responsible for both radical scavenging activities.

This correlation is particularly evident in the ranking of species efficacy, where *C. ellipsoidea* consistently demonstrated the lowest antioxidant capacity (DPPH• IC₅₀ = 395.39 ± 12.93 μg/mL; •NO IC_50_ = 577.10 ± 15.95 μg/mL), *C. vulgaris* showed intermediate performance (DPPH• IC_50_ = 361.79 ± 18.29 μg/mL; •NO IC_50_ = 523.96 ± 10.51 μg/mL), and *C. sorokiniana* exhibited the highest efficacy (DPPH• IC_50_ = 329.03 ± 4.30 μg/mL; •NO IC_50_ = 435.53 ± 10.20 μg/mL). This ranking pattern is consistently maintained across both assay systems, reinforcing the concept that similar antioxidant mechanisms are operating.

It is important to note that while the DPPH• assay measures the general free radical scavenging capacity against stable synthetic radicals, the •NO scavenging activity specifically targets nitric oxide radicals, which are predominantly generated during inflammatory processes. The similar efficacy patterns observed between these two distinct assays suggest that *Chlorella* extracts possess broad-spectrum antioxidant properties capable of neutralizing both general oxidative stress and inflammation-associated reactive nitrogen species. This dual functionality is particularly significant from a therapeutic perspective, as it indicates that these extracts may provide comprehensive protection against both general oxidative damage and inflammation-mediated nitrosative stress.

The consistent ranking across different radical scavenging assays is further validated by the FRAP results (Figure 4), where *C. sorokiniana* demonstrated the highest total antioxidant capacity (94.74 ± 5.72 mg TE/g), followed by *C. vulgaris* (66.21 ± 4.60 mg TE/g) and *C. ellipsoidea* (63.11 ± 6.23 mg TE/g). Although the FRAP ranking differs slightly from the DPPH• and •NO scavenging patterns, all three assays consistently identify *C. ellipsoidea* as having the lowest overall antioxidant performance, supporting the reliability of the observed trends.

Our findings align with previous studies on *Chlorella* species, which have extensively explored the in vitro and in vivo antioxidant activities of these microalgae. For instance, a previous report highlighted the peptide from *C. ellipsoidea* as an inhibitor of free-radical-induced oxidative stress [45]. In the case of *C. sorokiniana*, antioxidant enzymes including ascorbate peroxidase (APX), glutathione reductase (GR), glutathione S transferase (GST), peroxidase (POX), and superoxide dismutase (SOD) were presented [46]. Moreover, *C. sorokiniana* also demonstrates antioxidant properties, including studies involving a cell-based assay assessing the survival of *Caenorhabditis elegans* under oxidative stress [47], the inhibition of radical scavenging [48], the reduction in ROS products in the mitochondria [49], and reversible physiological oxidative perturbation [50]. Evaluation of the physiological response of *C. vulgaris* has revealed the presence of oxidative enzymes such as SOD and catalase (CAT) [51]. Furthermore, *C. vulgaris* has been investigated in vivo studies, including naphthalene-induced lipid peroxidation in the serum, liver, and kidneys of rats [52], and malondialdehyde (MDA), SOD, and glutathione peroxidase (GPx) in the livers of rats [53].

Despite the known presence of *n*-hexadecanoic and octadecanoic acid in *Chlorella* species, their antioxidant effects have been reported [54,55]. Previous studies have highlighted the antioxidant properties of hexanol, and its derivatives found in extracts from various sources such as pomegranate, mung bean, ripe coffee bean, and soybean [56,57,58]. These studies demonstrated that extracts with high concentrations of hexanol and its derivatives contained potent antioxidant compounds. The antioxidant activity observed in *Chlorella* extracts may be attributed to major constituents such as *n*-hexadecanoic and octadecanoic acid. Similarly, other algae species known to contain significant amounts of these fatty acids, such as *Spirulina platensis* and *Dunaliella salina*, have also demonstrated potential antioxidant activity [59,60]. Previous studies have indicated that algae’s fatty acid composition, including *n*-hexadecanoic and octadecanoic acid, contributes to their antioxidant properties [61]. This suggests that the antioxidant potential of algae like *S. platensis* and *D. salina* could be linked to their content of *n*-hexadecanoic acid and octadecanoic acid, supporting their role as natural sources of antioxidants. Therefore, our results support the potential antioxidant effects of *Chlorella* species. Notably, our findings indicated that the highest level of inhibition was observed with *C. sorokiniana*.

### 3.3. Antidiabetic Enzyme Inhibition by Chlorella Extracts

The α-glucosidase and α-amylase IC_50_ values of *Chlorella* extracts are presented in Figure 5 and Figure 6. Our results indicated that *C. sorokiniana* exhibited the highest potential α-glucosidase inhibitory activity (IC_50_ = 752.75 ± 57.95 μg/mL), followed by *C. ellipsoidea* (IC_50_ = 781.00 ± 104.51 μg/mL) and *C. vulgaris* (IC_50_ = 983.83 ± 24.20 μg/mL). The observed inhibitory effects were statistically significant for each treatment, except for 62.5 and 125 μg/mL of *C. sorokiniana*. Regarding the inhibition of α-amylase activity, it was noted that *C. sorokiniana* displayed the greatest inhibitory activity (IC_50_ = 3458.50 ± 104.01 μg/mL), followed by *C. vulgaris* (IC_50_ = 3677.77 ± 138.60 μg/mL) and *C. ellipsoidea* (IC_50_ = 4074.67 ± 288.50 μg/mL). Comparatively, the α-amylase and α-glucosidase inhibitory activities of *Chlorella* extracts were lower than those of acarbose (standard reference). The inhibitory effects exhibited concentration-dependent trends for different concentrations of the extracts and the positive control, acarbose. These results highlight the potential of *Chlorella* extracts, particularly *C. sorokiniana*, as inhibitors of α-glucosidase and α-amylase activities, which are crucial targets in managing diabetes.

Several research studies have explored the in vivo effects of antidiabetic agents [62,63]. However, the available literature is limited in terms of the specific effects of *Chlorella* extracts on the inhibition of α-glucosidase and α-amylase activities [36]. Some bioactive compounds have been identified for their ability to inhibit these enzymes. For example, hexanol from *Vitis vinifera* and *Agaricus campestris* has demonstrated hypoglycemic potential and insulin activity [64,65]. Notably, hexanol has not previously been associated with α-glucosidase and α-amylase inhibitors. On the other hand, previous studies on fatty acids and their effects on antidiabetic enzymes have shown that *n*-hexadecanoic and octadecanoic acids, identified as chemical constituents, exhibit the potent inhibition of α-glucosidase and α-amylase [41,42].

## 4. Conclusions

This study comparatively evaluated the phytochemical composition and bioactivities of lipid extracts from three *Chlorella* species (*C. ellipsoidea*, *C. sorokiniana*, and *C. vulgaris*), with GC–MS profiling revealing dominant fatty acids including *n*-hexadecanoic and octadecanoic acids across all species. Among the three species, *C. sorokiniana* KU.B2 demonstrated superior antioxidant activity (DPPH• IC_50_ = 329.03 ± 3.30 μg/mL; NO• IC_50_ = 455.53 ± 10.20 μg/mL) and the strongest antidiabetic potential (α-glucosidase IC_50_ = 752.75 ± 57.95 μg/mL; α-amylase IC_50_ = 3.458 ± 0.104 μg/mL), while *C. ellipsoidea* and *C. vulgaris* showed moderate activities.

Importantly, our study provides the first report of antioxidant and antidiabetic activities from the *C. sorokiniana* KU.B2 strain, highlighting its novelty and value as a local microalgal resource. The identification of bioactive constituents in this novel isolate opens new opportunities for the bioprospecting of microalgae from natural and agricultural environments. These findings establish a foundation for pharmaceutical development applications and demonstrate the significant potential of underexplored local *Chlorella* strains as sources of natural antioxidants and antidiabetic compounds for therapeutic applications.

## Figures and Tables

**Figure 1 biotech-14-00046-f001:**
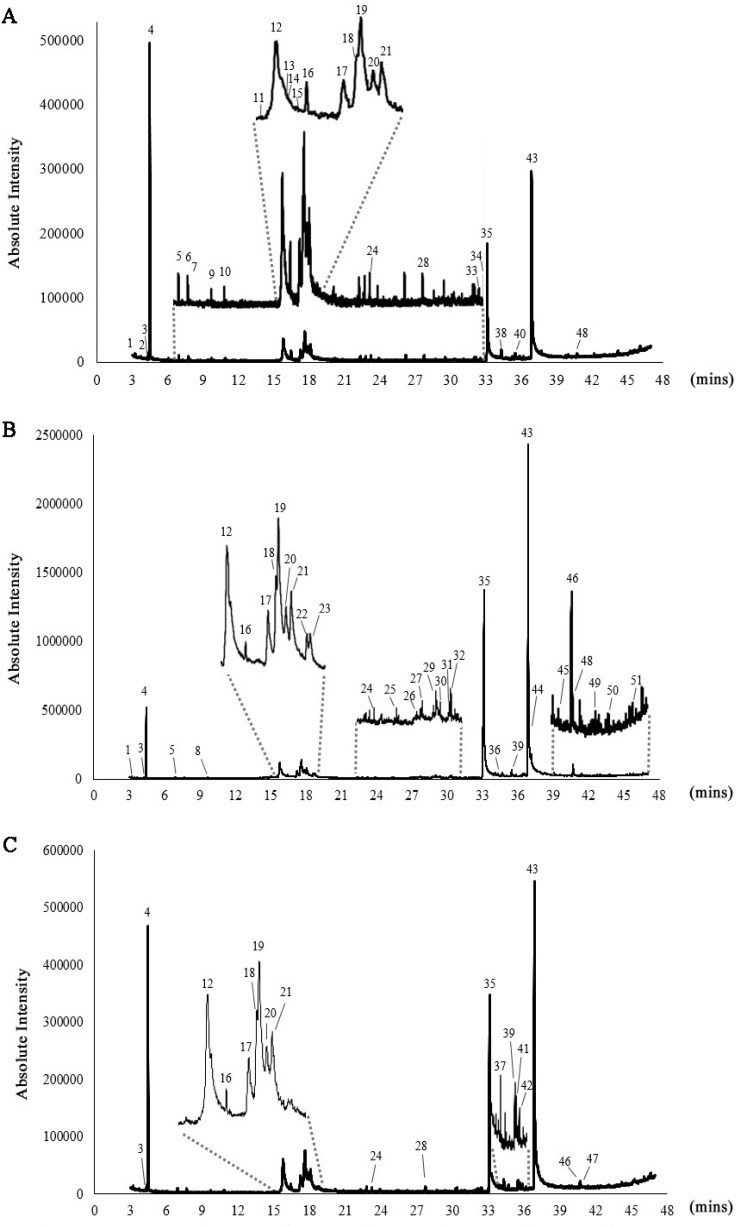
GC–MS chromatograms of *C. ellipsoidea* (**A**), *C. sorokiniana* (**B**), and *C. vulgaris* (**C**). The numbered peaks represent identified compounds, with their corresponding retention times (Rt) and relative abundances (%) provided in Table 1.

**Figure 2 biotech-14-00046-f002:**
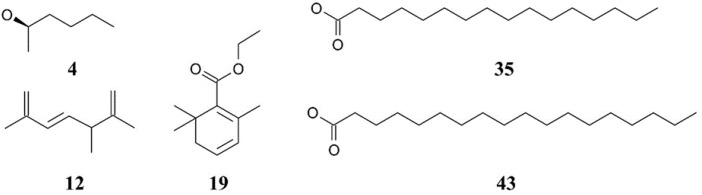
Chemical structures of the major compounds identified in *Chlorella* extracts. The compounds include 4—Butanal, 3-methyl- (isovaleraldehyde), 12—2,6-Octadiene, 2,6-dimethyl- (myrcene), 19—2,4-Hexadienoic acid, 3-(2,6,6-trimethyl-1-cyclohexen-1-yl)-, ethyl ester, 35—Hexadecanoic acid, ethyl ester (ethyl palmitate), and 43—9,12-Octadecadienoic acid (Z,Z)-, ethyl ester (ethyl linoleate). These compounds were identified using GC–MS analysis.

**Figure 3 biotech-14-00046-f003:**
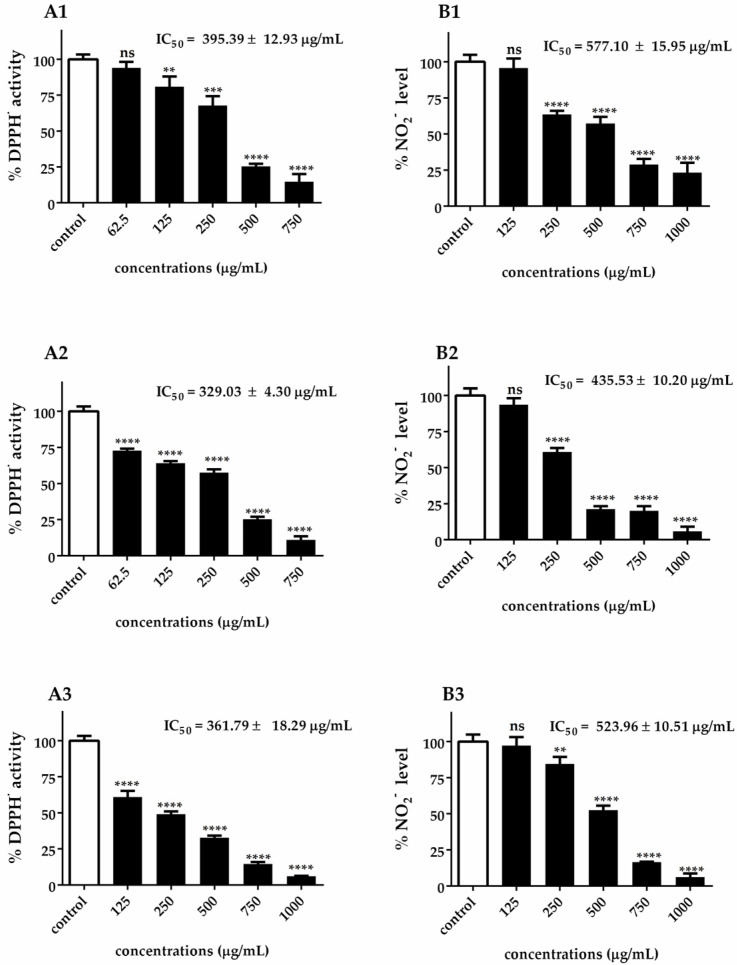
DPPH• radical scavenging activity (**A1**–**A3**) and •NO scavenging activity (**B1**–**B3**) of *C. ellipsoidea* (1), *C. sorokiniana* (2), and *C. vulgaris* (3) at various concentrations. Each value represents the mean ± SD (*n* = 3). IC₅₀ values are shown in each panel. Statistical significance was determined by two-way ANOVA followed by Tukey’s post hoc test for concentration-dependent effects within each species: *p* < 0.01 (**), *p* < 0.001 (***), *p* < 0.0001 (****), ns = not significant.

**Figure 4 biotech-14-00046-f004:**
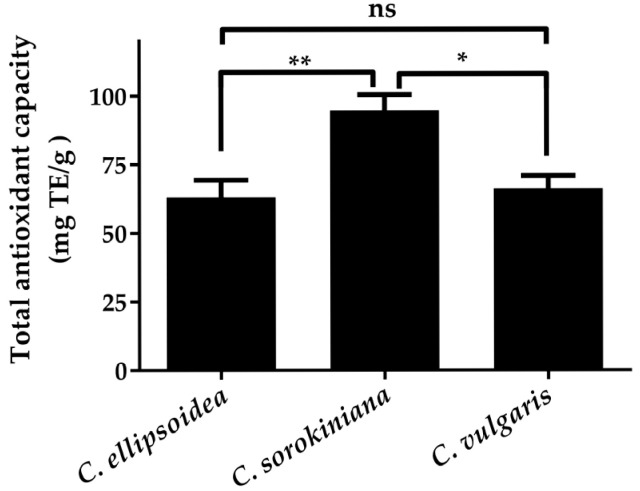
Total antioxidant capacity of *C. ellipsoidea*, *C. sorokiniana,* and *C. vulgaris* measured by FRAP assay. Data are presented as mean ± standard deviation (*n* = 3). Statistical differences were evaluated using two-way ANOVA followed by Tukey’s post hoc test. Significant differences: *p* < 0.05 (*), *p* < 0.01 (**), ns = not significant.

**Figure 5 biotech-14-00046-f005:**
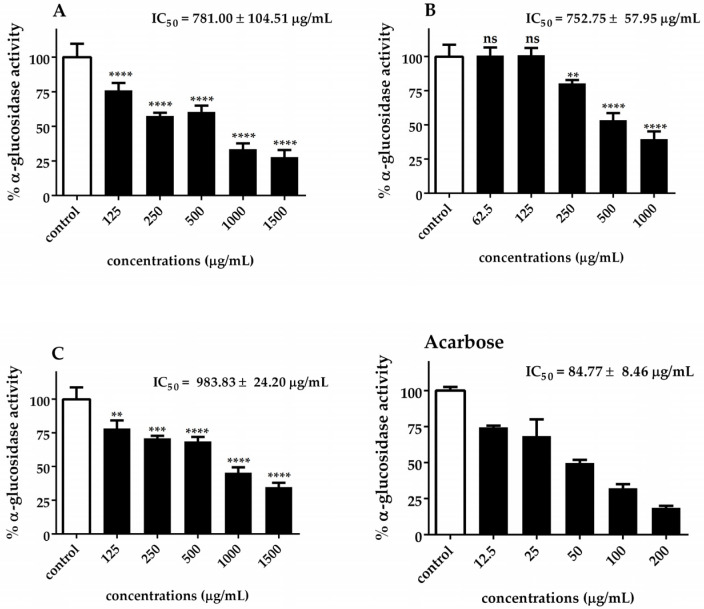
Inhibitory effect of *C. ellipsoidea* (**A**), *C. sorokiniana* (**B**), and *C. vulgaris* (**C**) extracts on α-glucosidase activity at various concentrations. Acarbose was used as a positive control. Data are expressed as mean ± SD (*n* = 3). IC₅₀ values are shown in each panel. Statistical significance was determined by two-way ANOVA followed by Tukey’s post hoc test: *p* < 0.01 (**), *p* < 0.001 (***), *p* < 0.0001 (****), ns = not significant.

**Figure 6 biotech-14-00046-f006:**
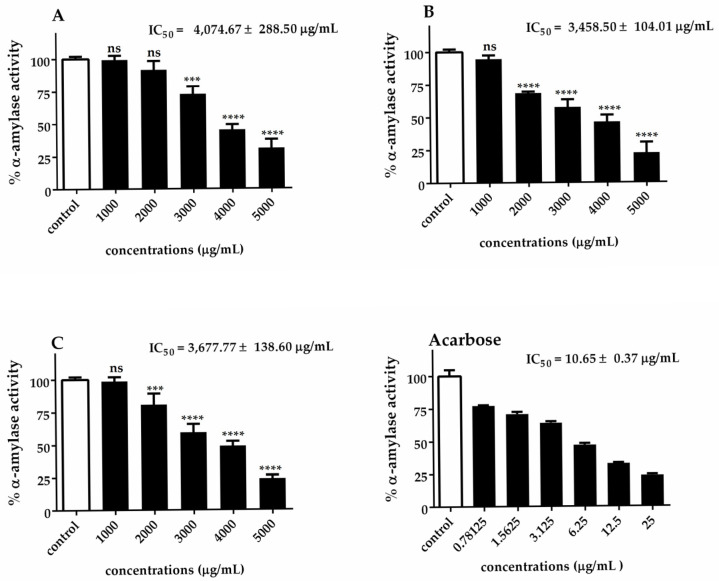
Inhibitory effect of *C. ellipsoidea* (**A**), *C. sorokiniana* (**B**), and *C. vulgaris* (**C**) extracts on α-amylase activity at various concentrations. Acarbose was used as a positive control. Data are expressed as mean ± SD (*n* = 3). IC₅₀ values are shown in each panel. Statistical significance was determined by two-way ANOVA followed by Tukey’s post hoc test: *p* < 0.001 (***), *p* < 0.0001 (****), ns = not significant.

**Table 1 biotech-14-00046-t001:** Chemical composition of *C. ellipsoidea*, *C. sorokiniana*, and *C. vulgaris* identified by GC–MS analysis. Compounds are listed with their retention time (Rt) and relative abundance (ratio) in each *Chlorella* extract. “—” indicates not detected.

No.	R_t_	Compound	Ratio		
			*C. ellipsoidea*	*C. sorokiniana*	*C. vulgaris*
1	3.237	3-Hydroxy-3-methyl-2-butanone	0.24	0.06	—
2	4.305	2,3-Dimethyl-1-hexene	0.24	—	—
3	4.422	Propanoic acid, 2-hydroxy-2-methyl-	0.43	0.08	0.31
4 *	4.475	2-Hexanol, (*R*)-	22.19	4.11	15.35
5	6.952	2-Butenal, 3-methyl-	0.38	0.07	—
6	7.724	Diisoamyl ether	0.35	—	—
7	7.773	4-Octen-3-one	0.22	—	—
8	9.702	Hexanal, 4-methyl-	—	0.06	—
9	9.74	Methacrolein	0.20	—	—
10	10.856	4-Heptanone, 3-methyl-	0.21	—	—
11	15.62	Carbamic acid, 3-methylphenyl-, butyl ester	0.37	—	—
12 *	15.815	1,3,6-Heptatriene, 2,5,5-trimethyl	6.23	2.64	5.96
13	16.035	Propanoic acid, 2,2-dimethyl-, 2-(1,1-dimethylethyl)phenyl ester	0.23	—	—
14	16.06	Methanol, (cyclohexyl)(2,3-dimethylphenyl)-	0.33	—	—
15	16.21	Carbamic acid, *N*,*N*-dimethyl-, 4-isopropylphenyl ester	0.19	—	—
16	16.469	*p*-Pentylacetophenone	0.79	0.14	0.37
17	17.268	5-Isopropyl-2,8-dimethyl-9-oxatricyclo[4.4.0.0(2,8)]decan-7-one	1.86	1.16	1.35
18	17.565	Cyclohexane, 1,1,4,4-tetramethyl-2,5-dimethylene-	2.20	1.17	2.45
19 *	17.644	4-(2,3-Dimethyl-2-cyclopenten-1-yl)-4-methylpentanal	6.16	3.53	6.26
20	17.907	1,3-Cyclohexadiene-1-carboxylic acid, 2,6,6-trimethyl-, ethyl ester	2.32	0.87	0.75
21	18.093	4-(1,2-Dimethyl-cyclopent-2-enyl)-butan-2-one	3.02	1.11	1.29
22	18.659	1-Cyclohexyl-2,2-dimethyl-1-propanol acetate	—	0.39	—
23	18.761	1H-Imidazolecarboxylic acid-, (1-methylethyl) ester	—	0.49	—
24	23.238	Pentanoic acid, 5-hydroxy-, 2,4-di-t-butylphenyl esters	0.43	0.11	0.34
25	25.319	3,4,5-Trimethyl-4-heptanol	—	0.10	—
26	27.579	1-Tridecyn-4-ol	—	0.06	—
27	27.738	Dodecane, 2,6,11-trimethyl-	—	0.11	—
28	27.743	3,5-Dimethyl-4-octanone	0.48	—	0.30
29	29.001	Tetradecanoic acid	—	0.17	—
30	29.381	1,4-Benzenedimethanol, α,α′-dimethyl-	—	0.08	—
31	30.246	Cyclohexanepropanol-	—	0.05	—
32	30.38	Isopropyl myristate	—	0.18	—
33	32.151	3-Hexanone, 2,5-dimethyl-	0.22	—	—
34	32.995	Phthalic acid, 4-cyanophenyl nonyl ester	0.46	—	—
35 *	33.112	*n*-Hexadecanoic acid	17.81	33.30	23.55
36	34.025	Glutaric acid, tridec-2-yn-1-yl 4-methylcyclohexyl ester	—	0.11	—
37	34.307	Hexadecanal	—	—	0.84
38	34.314	14-Heptadecenal	0.69	—	—
39	35.476	*n*-Pentadecanol	—	0.85	0.86
40	35.492	1-Undecene, 9-methyl-	0.62	—	—
41	35.618	7-Tetradecyne	—	—	0.63
42	35.847	2-Heptadecenal	—	—	0.45
43 *	36.868	Octadecanoic acid	30.70	46.39	38.14
44	37.127	Hexadecanamide	—	0.59	—
45	39.563	2-Propenoic acid, 3-(4-methoxyphenyl)-, 2-ethylhexyl ester	—	0.08	—
46	40.668	Octadecanamide	—	1.08	0.50
47	40.675	9-Octadecenamide, (*Z*)-	—	—	0.28
48	40.725	Hexanedioic acid, bis(2-ethylhexyl) ester	0.42	0.50	—
49	42.713	1-Dimethylaminohexane	—	0.13	—
50	43.756	1-Pentadecyne	—	0.10	—
51	45.772	Phthalic acid, di(3-methylphenyl) ester	—	0.11	—

Note: * indicates the major compounds identified in *Chlorella* extracts.

## Data Availability

The original contributions presented in this study are included in the article. Further inquiries can be directed to the corresponding author.

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
