# Peer review of "In Vitro Antioxidant Potential, Antidiabetic Activities, and GC–MS Analysis of Lipid Extracts of Chlorella Microalgae"

_biotech, 2025, doi:10.3390/biotech14020046_

Round 1
Reviewer 1 Report
Comments and Suggestions for Authors
The manuscript entitled “In Vitro Antioxidant Potential, Antidiabetic Activities, and GC-MS Analysis of Lipid Extracts of Chlorella Microalgae” is an interesting piece of work that demonstrates the potential impact of bioactive compounds from three microalgae. However, the quality of the manuscript can be strengthened by addressing the following queries:
- In the introduction section, the authors should provide justification for selecting three microalgae in the present study.
- If the selected algae are cited in references 15-28, what is the incremental advancement made in this study? If the microalgae are unexplored, please mention this novelty in the abstract, at the end of the introduction, and in the conclusion sections.
- Please discuss the lipid content of the selected microalgae on a dry weight basis for better presentation.
- It would be interesting if the authors could comment on the specific roles of chemical constituents in antioxidant and antidiabetic applications.
- In Figure 2, please highlight the percentage of major compounds identified in Chlorella extracts to explore their potential applications.
- The authors should discuss why the patterns of results are similar for DPPH radical scavenging activity (A1-A3) and NO scavenging activity (B1-B3), as shown in Figure 3.
- The references need to be updated to include recent studies from the last two years (2024-2025).
Reviewer 2 Report
Comments and Suggestions for Authors
I reviewed the manuscript entitled “In vitro Antioxidant Potential, Antidiabetic Activities, and GC-MS Analysis of Lipid Extracts of Chlorella Microalgae”, submitted to the journal BioTech. The manuscript is well written and clearly presented, and it addresses a relevant topic within algal biotechnology. The authors compare three species of Chlorella (C. ellipsoidea, C. sorokiniana, and C. vulgaris) in terms of lipidomics, antioxidant activity, and in vitro antidiabetic potential. The results appear reproducible, but I would like to suggest a few revisions before recommending the manuscript for publication in BioTech.
- Introduction
- In the introduction, the authors describe Chlorella as "promising sources." However, in my opinion, Chlorella is already a well-established platform for algal bioproducts, being one of the most widely cultivated microalgae globally. To improve the context of this paragraph, I suggest reframing the narrative to focus on the development of new bioproducts derived from Chlorella. See, for example:
- https://doi.org/10.1371/journal.pone.0255996
- https://doi.org/10.1016/j.ijbiomac.2024.138630
-
- The originality of the study should be clearly stated, and the objectives should explicitly mention the comparative analysis between the three species.
- Materials and Methods
Some adjustments are needed to ensure reproducibility:
-
- Lines 95–96: Please indicate the harvesting (e.g., centrifugation or flocculation) and drying methods (e.g., lyophilization or oven drying), as these processes can significantly affect biomass quality.
- Lines 156–159: Specify which type of ANOVA was used and confirm whether assumptions for parametric testing were checked.
- Results
- Figure 3 caption: The authors state that significant differences were assessed using two-way ANOVA followed by Tukey’s post hoc test. However, they only report the p-values related to concentration differences. Since a two-way ANOVA was performed, I suggest grouping the bars by factor (i.e., a single graph for DPPH and another for NO₂⁻) to better visualize species-related differences. Also, include letter notations to indicate statistically significant differences (use lowercase letters for comparisons across concentrations within the same species and uppercase letters for comparisons across species at the same concentration).
- The same comment applies to Figures 5 and 6.
- Conclusion
The conclusion should be rewritten to be more direct and clearly aligned with the study’s objectives, similar to the clarity found in the abstract.
Reviewer 3 Report
Comments and Suggestions for Authors
The work presented a study of the composition and biological activities of lipid extracts from three Chlorella microalgae species: C. ellipsoidea, C. sorokiniana, and C. vulgaris. Although not very innovative, the study is well-conducted and relevant to the scientific field of natural product research with a strong focus on biochemistry and pharmaceutical applications.
The language is correct and the introduction is generally informative and does a good job stating the research question.
Several issues to be addressed before publication:
Major issues:
Line 98 -7 days seems like a very long and unusual time for extraction. Please add a reference or include extraction optimization
Line 181-204-those paragraphs should be reduced or removed since they are not relevant to the composition or bioactivities under study.
Please include in the discussion at least an attempt to explain the differences in the bioactivity of the chlorella species based on the differences in the lipid profile determined in the study.
I suggest replacing DPPH and NO with DPPH• and •NO
Figure 3- These results are normally presented as % inhibition, not as % activity, which is a bit confusing.
How was the IC 50 calculated? There are several approaches to this calculation: interpolation, mathematical modeling etc https://doi.org/10.1002/pst.426, please state your methodology in the materials and methods section.
Minor
Line 40- DNA does not need to be explained, is considered a word. https://www.merriam-webster.com/dictionary/DNA
Line 91-define TAP
.Line 124 – add concentration and pH to the buffer description. Same in line 140
Line 136- replace line equation by the calibration curve
Line 170-172 - please specify if in the cited bibliography they found the same major lipidic molecules.
Line 176 –“from C16 to C18 (> 92%)18.” The last 18 is a typo?
Round 2
Reviewer 2 Report
Comments and Suggestions for Authors
The authors have adequately addressed all the concerns I raised during the first round of peer review.
Author Response
Comment: The authors have adequately addressed all the concerns I raised during the first round of peer review.
Response: Thank you for your valuable comments and suggestions.
Reviewer 3 Report
Comments and Suggestions for Authors
Thank you for the clear answers; all my queries were answered.
I would like to request to review the manuscript for differences in the nomenclature for DPPH• and •NO, which is still incorrect in several places, e.g., lines 264-271 and 294.
Author Response
Comment 1: Thank you for the clear answers; all my queries were answered.
Response 1: Thank you for your valuable comments and suggestions.
Comment 2: I would like to request to review the manuscript for differences in the nomenclature for DPPH• and •NO, which is still incorrect in several places, e.g., lines 264-271 and 294.
Response 2: We agree, and the issues raised by the reviewer have been addressed throughout the manuscript (Lines 148-153, 160, 266-271, 282-283, 295, 331 and highlighted in yellow).